# Preliminary Study on Microbial Deterioration Control and Effectiveness Evaluation in the Neolithic Prehistoric Archaeological Site of Dadiwan, Northwest China

**Ruihong Xu [1,2,†], Yuxin Chen [1,2,†], Dongpeng He [1,2], Guobin Zhang [1,2,*], Qiang Luo [3], Hongtao Zhan [1,2] and Fasi Wu [1,2,4,*]**

[1] National Research Center for Conservation of Ancient Wall Paintings and Earthen Sites, Department of Conservation Research, Dunhuang Academy, Dunhuang 736200, China; xrh50061@dha.ac.cn (R.X.); cyx20431@dha.ac.cn (Y.C.); hedp456@163.com (D.H.); zht13239496273@163.com (H.Z.)

[2] Gansu Provincial Research Center for Conservation of Dunhuang Cultural Heritage, Dunhuang Academy, Dunhuang 736200, China

[3] The Museum of the Terracotta and Horses of Qin Shihuang, Xi'an 710600, China; bmyluoqiang@126.com

[4] Key Laboratory of Extreme Environmental Microbial Resources and Engineering, Northwest Institute of Eco-Environment and Resources, Chinese Academy of Sciences, Lanzhou 730000, China

\* Correspondence: zhanggb@dha.ac.cn (G.Z.); wufs@dha.ac.cn (F.W.)

† These authors contributed equally to this work.

**Abstract:** Microbial deterioration as one of the widespread problems in archaeological site museums significantly affects their safety and exhibits. This paper systemically investigated the environments and conditions of microbial outbreaks in the Dadiwan No. F901 site museum, which is a representative archaeological site of prehistoric Yangshao culture. The morphology and harmful characteristics of the outbreak microorganisms were analyzed by microscopic techniques. The ultraviolet resistance of harmful microorganisms was also studied. Combining these findings with the original facilities of the site museum, a scientific and reasonable project was proposed to control and prevent the activity of harmful microorganisms. In addition, a 1% OIT/DCOIT biocide concentration was applied to inhibit microorganism-caused deterioration, in combination with mechanical removal based on laboratory tests and screening in situ. The effectiveness of microbial control was assessed using a portable microscope, ATP bioluminescence assay, and color difference detection. As a long-lasting preventive measure for microbial deterioration, an ultraviolet sterilization system can efficaciously prevent the re-outbreak of microbial deterioration to form a relatively stable dynamic balance for the surroundings of the site. This study is a resultful exploration in terms of microbial control and plays an important role in the sustainable protection of archaeological site museums.

**Keywords:** Dadiwan site; microbial deterioration; microbial elimination; environmental control; prevention and conservation





## 1. Introduction

The construction of a site museum is a primary method for the in situ conservation of cultural sites [1–3]. However, in situ conservation remains challenging work influenced by various factors, including climate, environment, and the vulnerability of relics. Among them, the outbreak and contagion of harmful microorganisms, particularly prominent in environments with high relative humidity [4], have seriously negative impacts and exacerbate the deterioration process of sites in site museums. This poses a disadvantage for the conservation and exhibition of historical information in cultural sites.

In the early 1980s, mold deterioration broke out on the murals in the Takamatsuzuka Tomb of Japan due to irreversible changes in indoor environmental conditions, high-temperature and humidity environments, and the aging of restorable materials. The deterioration was so severe that the Takamatsuzuka Tomb had to be dismantled and

relocated after 2000 years [5]. The prehistoric rock paintings in the Lascaux Cave in France have been subjected to erosion by various microorganisms since discovery to date. Bastain F et al. have reported that the main factors contributing to the microbial outbreak were large-scale archaeological excavations and numerous tourists [6]. Similar to the Lascaux Cave, Altamira prehistoric rock art was forced to close because of microbial erosion [7,8].

As relevant studies about heritage conservation are gradually being produced by experts, it is suggested that microorganisms are one of the vital factors detrimental to the conservation and sustainability of heritage sites. Currently, microbially induced calcite precipitation as a preservation method has been applied in earthen sites for site consolidation [9,10]. However, in recent years, microbial deterioration has been widely threatening the preservation and display of earthen sites. It has been reported that the organic acids produced by microorganisms could react with metal ions to crystalize on the surface of sites, leading to cracks, efflorescence, desquamation, etc. Meanwhile, the mycelium from fungi could penetrate the substrate of the site to result in cracks that impact stability [11,12]. The diversity and distribution of microorganisms have been researched in many earthen sites [13–15], and related control and preventive measures have been proposed to handle microbial deterioration [16–20], but they are rarely put into practice in large-scale conservation projects of earthen sites as study achievements.

The Dadiwan Neolithic site is located at the east of Shaodian Village, Wuying Town, 45 km northeast of Qin'an County, Gansu Province, Northwest China. It dates back approximately 7800 to 4800 years and covers a total area of 2.75 million square meters. The discovery of the site is of great significance for establishing the sequence of prehistoric culture in the upper reaches of the Wei River, studying the emergence and development of neolithic culture in the Yellow River basin, and exploring the historical process of the origin of Chinese civilization, especially the No. F901 palace site, which represents the greatest achievements of Shaoyang cultural buildings [21,22]. Since 2018, after the construction of the new site museum building and the installation of an internal glass curtain wall, the site has been in a relatively closed state of preservation. In 2019, the site managers found white colonies on the surfaces of the site and surrounding buildings where the amounts of microbial colonies had been seen to increase and extend to threaten the long-term preservation of the site over time. Therefore, the aim of this study is to propose a complete and long-term effective technique process and conservation project to solve the current microbial deterioration problems existing in the Dadiwan F901 site museum and provide scientific support for the construction and sustainable management of site museums in the future.

## 2. Materials and Methods

### 2.1. Study Site and Environmental Monitoring

The Dadiwan No. F901 site museum is located in Yanjiagou Village, at the southwest corner of the Dadiwan Neolithic Site Protection area (35°0′45.9″ N, 105°55′10.2″ E), with an altitude of about 1673 m. The F901 site covers an area of about 420 square meters and is a well-preserved multi-room complex. The layout of the entire site is regular and symmetrical. The site sets a precedent for later palace architecture as the largest magnificent building discovered so far in the archaeological exploration of the Neolithic age in China. Notably, the earliest concrete floor and the earliest building of fire-prevention technology fully reflect the superb wisdom and exquisite skills of our ancestors over 5000 years ago. In 1988, the Dadiwan F901 site was designated as a National Key Cultural Relic Protection Unit [23,24].

In this study, environmental monitoring was specifically conducted to explore the relationship between microbe outbreaks and relevant environmental factors. Temperature and relative humidity (RH) outside the site museum were continuously monitored hourly using a HOBO® U-30 data logger (Onset Computer Corporation, Bourne, MA, USA) installed in the meteorological station and another HOBO® U23-001 data logger (Onset Computer Corporation, Bourne, MA, USA) attached to the top of the middle of the north

wall of the F901 site. Temperature, RH, and air dew point temperature were automatically recorded at intervals of 10 min. Detailed monitoring locations are shown in Figure 1.

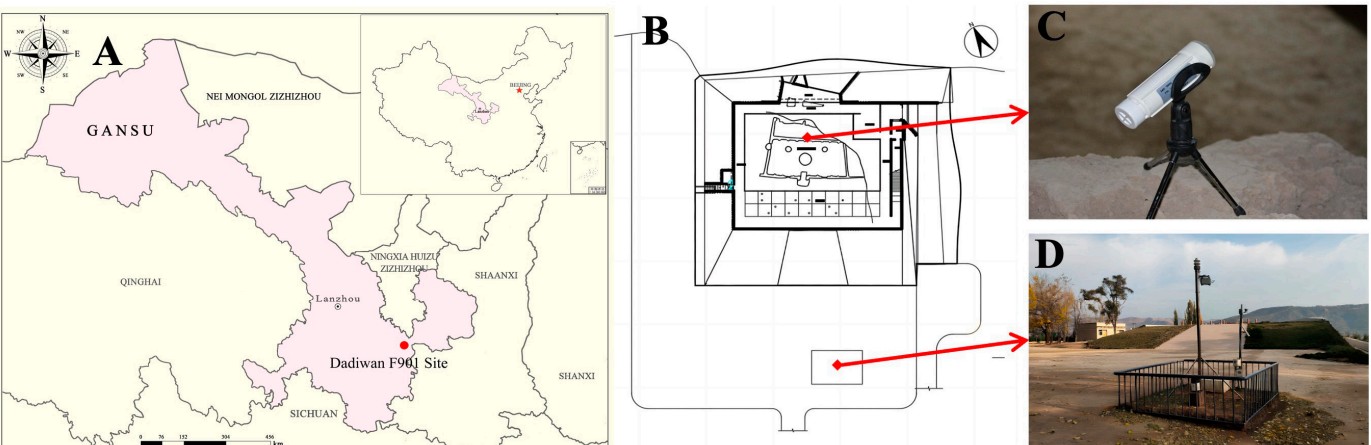

**Figure 1.** The location of the Dadiwan F901 site museum (**A**) and detailed monitoring site (**B**) at the inner of the F901 site (**C**) and outside the Dadiwan site museum (**D**).

### 2.2. Morphology of Harmful Microorganisms

#### 2.2.1. In Situ Observation

Representative locations were selected on the surfaces of site F901, including the main site (walls, exterior walls from the pillar hole, and ground), backfill soil and cement outside the site, surface of the glass curtain wall, and surroundings. A portable microscope (3R-MSA620WF, Anyty, Beijing, China) was used to observe microbial growth in the field at these locations and capture pictures for the record.

#### 2.2.2. Profile Characteristics of Samples

Small particle residues attached to the obvious microbial colonies were collected from the debris of the site ground and cement ground of peripheral buildings. To preserve structural relationship information between the microorganisms and the site surfaces, these residues were embedded in situ using epoxy resin and a curing agent (Epo Thin TM, Buehler, Lake Bluff, IL, USA). Subsequently, they were transferred to the laboratory for polishing with a polisher (BUEHLER metaserv 250, Lake Bluff, IL, USA) for microbial observation with a polarizing microscope (DM2200P, LEICA, Wetzlar, Germany) to analyze the growth situation and the interaction between microbial filaments and soil [25].

### 2.3. Removal Methods of Harmful Microorganisms

In line with the characteristics of the soil structures and materials, microbial attachment, and microenvironmental conditions at the Dadiwan F901 site, a combination of mechanical removal and biocides was utilized to eliminate microorganisms for site preservation. A negative-pressure dust suction device equipped with various types of soft brushes and other mechanical cleaning methods was used to eliminate microbial colonies and residues from the surfaces of the site [26]. Subsequentially, a biocide was applied to inhibit microorganisms, targeting microbial spores and filaments in the pores or cracks of the site soil.

#### 2.3.1. Screen of the Mechanical Removal Tools

Dadiwan F901 site, constructed from sandy soil, features a coarse surface with numerous cracks and loose wall structures, creating an environment where detrimental microorganisms are widely distributed. Consequently, the use of sharp tools, such as metal tweezers and scalpels, is impractical. Considering the growth characteristics of harmful microorganisms and the surface structure characteristics of the site, three kinds of cotton

swabs and seven kinds of brushes with different materials and shapes were selected as mechanical removal tools for in situ experiments. This approach aimed to determine more suitable tools for the site's condition. Besides mechanical removal, modified negative-pressure dust-removal equipment capable of creating a small range of negative-pressure environments (−21 KPa) was employed to effectively remove diffuse microorganisms in the air, meeting the sterilization requirements within the construction environment of the site museum (Figure 2).

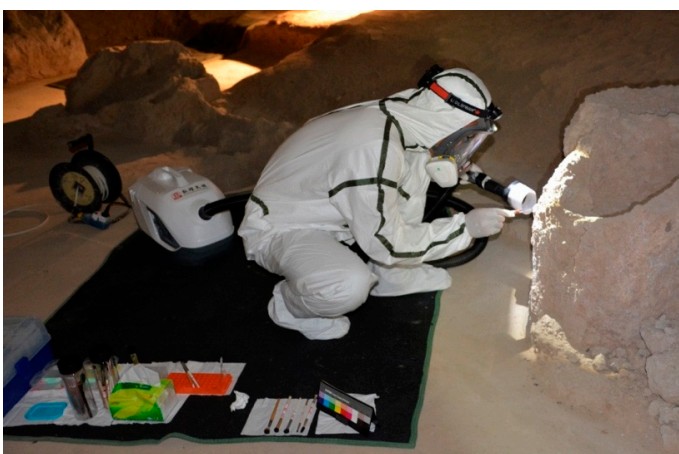

**Figure 2.** Testing the in situ removal efficiency of harmful microorganisms distributed on the site surfaces using different tools.

2.3.2. Screen of Chemical Biocides in the Laboratory and In Situ

In this study, five kinds of biocides, MD (Dimethyl Fumarate) [27], BC (Benzalkonium Chloride) [28–30], OIT (2-octyl-4-isothiazolin-3-one), DCOIT (4, 5-dichloro-N-octyl-4-isothiazolin-3-one) [26,31], and EUXYI K100 (active ingredient is methyl chloroisothiazolinone, hereinafter called K100) [28,32], were selected to examine the elimination effectiveness of the samples which were collected from the Dadiwan F901 site using a bacteriostatic zone experiment. For the experiment, 0.5% concentrations of the aforementioned biocides and a 0.5% OIT/DCOIT compound were prepared as the experimental groups. In contrast, 75% ethanol and ddH$_2$O were prepared as the control and blank group, respectively. In the bacteriostatic zone experiment, approximately 0.1 g of collected soil samples with microbial filaments was put into centrifuge tubes with 1 mL of sterilized water. After vortexing for 10 min, the supernatant was collected and diluted to $10^{-1}$. The diluted microbial solution was then spread on an R2A solid medium (yeast extract 0.5× $g$, peptone 0.5× $g$, casein hydrolysate 0.5× $g$, glucose 0.5× $g$, soluble starch 0.5× $g$, dipotassium hydrogen phosphate 0.3× $g$, and DDH$_2$O 1000 mL) [27]. Circular drug susceptibility papers with a diameter of 6 mm were placed on the solid medium, and 10 μL of the prepared biocide solution was applied when the microbial solution dried on the solid medium. There were three pieces of drug susceptibility paper placed on each plate as triplicates. The experimental plates were incubated at room temperature for 7 days for observation.

To examine the effective concentration of chemical biocides, typical soil samples from the site were selected as follows: the coarse mud layer on the west side inside the exterior northern wall (A), the smooth mud layer on the east side outside the exterior northern wall (B), the ground of the southern side (C), and the cement wall of the northern side peripheral building N6 (D), shown in Figure 3. The test areas were divided into four test parts, each with a side length of 10 cm × 10 cm. OIT, DCOIT, OIT/DCOIT compound, and K100 with concentration gradients of 0.25%, 0.5%, 0.75%, and 1.0% were used as experimental treatments [33]. Distilled H$_2$O and 75% ethanol were regarded as a blank group and control group, respectively. The treatment of each test area was carried out according to the operation process of the first mechanical removal, an initial spraying by

the biocide and an identical second spraying as the second step. A nebulizer was used to spray the corresponding type and concentration of biocides in each test area, and it was continuously applied twice with an interval of 48 h. Each time, approximately 1.7 mL of biocide was sparged to penetrate around 2 mm depth from the surface.

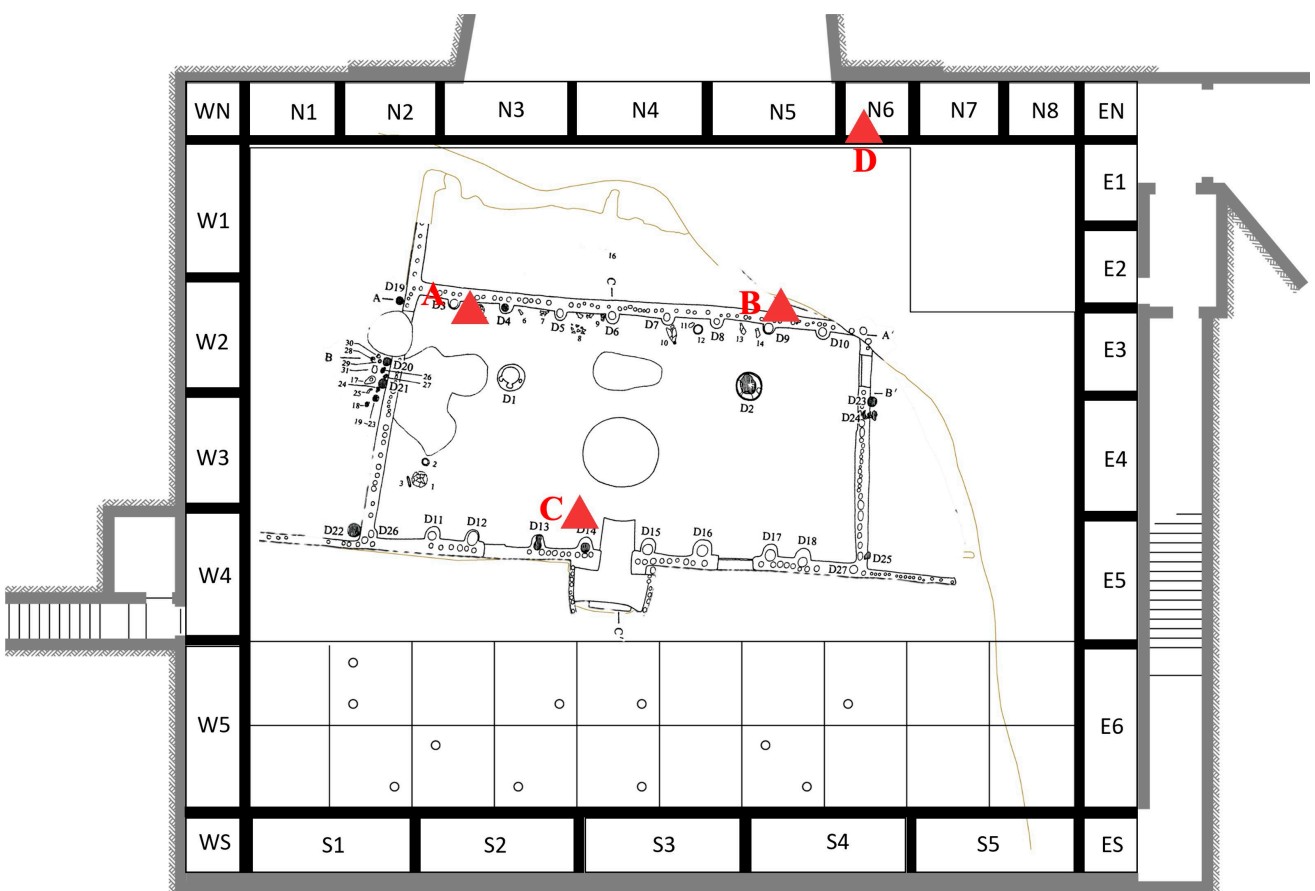

**Figure 3.** The test points for efficiency measurement of biocides with different concentrations. (A) The coarse mud layer on the west side inside the exterior northern wall, (B) the smooth mud layer on the east side outside the exterior northern wall, (C) the ground of the southern side, and (D) the cement wall of the northern side peripheral building N6.

*2.4. Inhibition of Ultraviolet Light (UV Light) to Harmful Microorganisms*

2.4.1. Inhibition of UV Light to Culturable Microorganisms

Culturable fungi and bacteria isolated from the F901 site were used to test the resistance to UV light. Approximately 0.1 g of soil samples with filaments was added to a 5 mL centrifuge tube with 1 mL of sterilized distilled water. The samples were vortexed for 10 min and the supernatants were then transferred and diluted to $10^{-4}$ using sterilized distilled water in 1.5 mL tubes. A 200 μL solution was evenly spread on a PDA medium (potato $200\times g$, glucose $20\times g$, agar $15\times g$, and ddH2O 1000 mL) and R2A medium; PDA and R2A are commonly utilized to isolate microorganisms that grow on cultural heritage sites [27,33]. All the inoculated plates were placed on a clean bench and individually irradiated by UV light for varying durations: 0 min, 30 min, 60 min, and 120 min, as illustrated in Figure 4A.

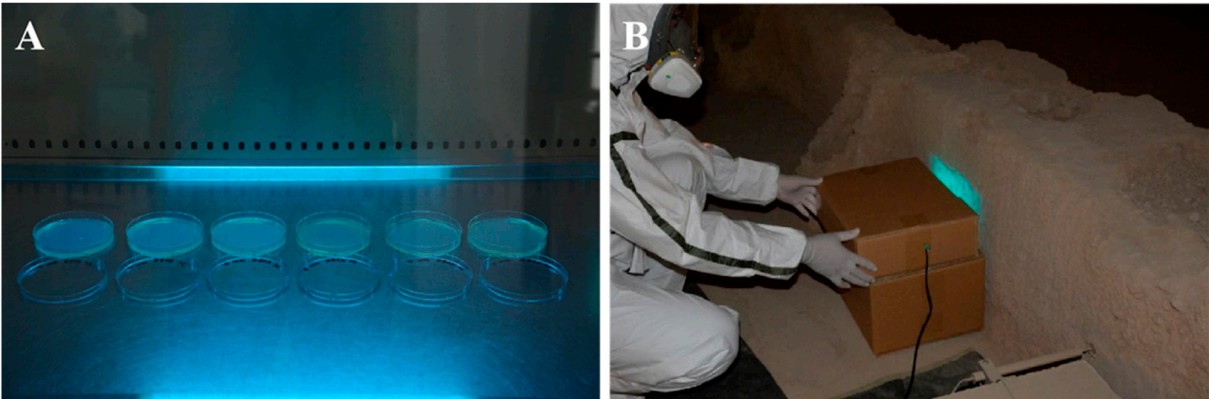

**Figure 4.** Experiment of UV light irradiation for inhibition of microorganisms. (**A**) Microbial inhibition by UV light at the laboratory; (**B**) field examination of the UV light inhibition effect on the microorganisms on site surfaces.

### 2.4.2. Test of Microbial Inhibition by UV Light In Situ

The wall of the site and cement walls of peripheral buildings were selected as experimental objects because the surfaces of these walls are nearly smooth, with uniformly distributed microbial colonies. A simple UV light box was made, with aluminum foil attached to the inner wall. In the box, a 16 W lamp was installed with a distance of 40 cm from the open surface of the carton, which was 15 cm × 42 cm. This simple box was aligned to the tested point and gently leaned against the wall, as in Figure 4B. The irradiation time was set for four periods of 0 min, 60 min, 90 min, and 180 min, based on the impact of the coarse surfaces of the site walls on the effectiveness of UV light irradiation.

### 2.5. Evaluation of Removal Effect on Harmful Microorganisms

### 2.5.1. Microscopic Observation

The morphologies of sample areas, which were treated by mechanical removal, biocides, and UV light, were observed and recorded by a portable microscope. For each treatment, the effects of microbial elimination before and after treatment needed to be observed and recorded for evaluation.

### 2.5.2. ATP Bioluminescence Assay

ATP bioluminescence assay used an ATP bioluminescence detector (Lumitester TM PD-30, Kikkoman, Noda, Japan) and pen (LuciPacTM Pen, Kikkoman, Noda, Japan) for detection. For sampling, after a sterilized swab was soaked in sterilized water for 4 s, it was applied to slightly dip the surface of a 2 cm × 2 cm area for microbial collection. The swab for sampling was put into a swab tube and shaken sufficiently to completely solubilize the sample on the swab with cell lysate and fluorescent compound. The amount of fluorescence could be obtained after 10 s when the swab tube was inserted into the bioluminescence detector. The eliminated ratios were calculated by the following formulas, respectively:

$$\text{For mechanical removal: eliminated ratio (\%)} = [(C_0 - C)/C_0] \times 100\% \tag{1}$$

$$\text{For biocide elimination: eliminated ratio (\%)} = [(C - C_1)/C] \times 100\% \tag{2}$$

where $C_0$ is the value of ATP fluorescence before treatment. C represents the value of ATP fluorescence after treatment by mechanical removal. $C_1$ is the value of ATP fluorescence treated by biocides.

### 2.5.3. Color Difference Detection

Color change of the surface with microbial deterioration was detected using a colorimeter (NR60CP, 3NH, Shenzhen, China). The color difference value was obtained by

detecting the color difference value before and after treatments by mechanical removal, chemical biocides, or UV light, using the following formula:

$$\Delta E^*_{ab} = ((\Delta L^*)^2 + (\Delta a^*)^2 + (\Delta b^*)^2)^{1/2} \tag{3}$$

where $L^*$ stands for lightness, $a^*$ is the red–green parameter, and $b^*$ is the yellow–blue parameter. $\Delta L^*$, $\Delta a^*$, and $\Delta b^*$ were calculated from the formulae $\Delta L^* = L^*_{before} - L^*_{after}$, $\Delta a^* = a^*_{before} - a^*_{after}$ and $\Delta b^* = b^*_{before} - b^*_{after}$ ("before" represents the values before the treatments or before biocide treatment when treatments of mechanical removal finished, and "after" stands for the values after the treatment of mechanical removal or after two-step treatment with biocides, respectively). To reduce the measurement error, nine measurements were made in the range of this experimental area for each test. The results were the average of the nine measurements.

## 3. Results and Discussion

### 3.1. Changes in Environmental Factors

During the one-year monitoring period, the temperature of the external surroundings fluctuated with the seasons from −14.31 °C to 34.86 °C. The annual mean temperature was 9.25 °C. The monthly average temperature reached its lowest point in December 2019 at −2.38 °C and peaked in August 2019 at 19.35 °C. The annual RH varied from 5.5% to 100%. The annual average RH was 74.5%. The monthly average RH was the lowest in March 2020, while the highest occurred in September 2019.

In comparison to the temperature of the external surroundings, the ambient temperature inside the F901 site experienced a slight fluctuation throughout the seasons, ranging from 4.92 °C to 20.32 °C, with an annual average temperature of 12.48 °C. The highest and lowest monthly average temperatures were recorded in August 2019 and February 2020, respectively. The annual RH was from 63.5% to 100%, with an annual average RH of 97.1%. Additionally, the monthly average RH consistently exceeded 80% (Figure 5).

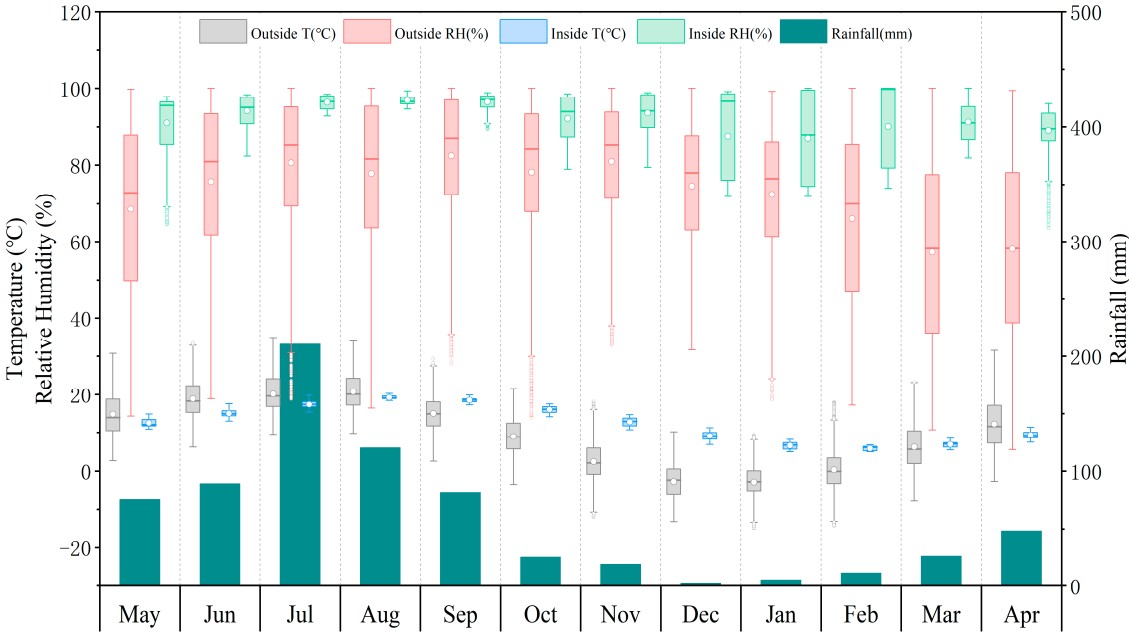

**Figure 5.** Box plot of temperature and relative humidity inside and outside of Dadiwan F901 site and column chart of rainfall capacity from May 2019 to April 2020.

In recent years, some Chinese site museums have also encountered biodeterioration problems. For example, shortly after the Tanjiapo Relic Museum of Tongguan Kiln in Changsha, Hunan Province was built, a huge number of plant roots remained, which

became a source of saprophytic microorganisms under the catalysis of high temperatures and high humidity, leading to an outbreak of mold deterioration [34]. The Shanghai Zhidanyuan Yuan Dynasty Sluice Site Museum is characterized by a semi-open building with prominent fluctuations in indoor environmental temperature and humidity, so the surfaces of the earthen and wooden structures exhibit shrinkage, cracking, and microbial activities in the frequent state of a dry and wet cycle [35]. In the early excavations of No. 1 pit of Qin Shihuang's Terracotta Warriors and Horses in Xi'an, Shaanxi Province, mold deterioration broke out because of high temperatures, humidity, and a mass of tourist activities. Consequently, the quantity of microorganisms in the indoor environment of the Qin Mausoleum has been a significant environmental concern for cultural relic experts [36–38]. As for the Hanyang Mausoleum Museum, the burial pit site is connected to the ground and completely enclosed by the museum buildings, resulting in continuous upward evaporation of underground water with 100% relative humidity indoors throughout the year, thereby aggravating the growth of mold [39,40]. Hence, it is evident that environmental factors have a great impact on microbial growth and biodeterioration to cultural heritage sites.

### 3.2. Morphological Characteristics of Harmful Microorganisms
3.2.1. In Situ Morphology and Distribution of Harmful Microorganisms

In this study, the inner environment of the Dadiwan F901 site museum was systemically investigated (Figure 6A). According to the field investigation, the microbial colonies on the wall surfaces of the Dadiwan F901 site could be visible to the naked eye and displayed white filaments, part of which were mixed with soil dust particles or white particles, under microscopic observation (Figure 6C, D). There were no visible white filaments on the ground of the site and backfill area outside the main site (Figure 6E,F), but the microorganisms covered by dust on the surfaces could be observed with a portable microscope, which has been widely applied in the conservation of heritage sites with the advantages of non-destruction, efficiency, simple operation, etc., for microbial observations, especially when it is challenging to assess the deterioration situation on the surfaces of a site when the microorganisms could not develop into complete colonies observable by the naked eye [41–44]. Not only can the portable microscope observe in situ for quick diagnosis of situations caused by microbial deterioration, but it also generates archival data through pictures. These data help evaluate cultural protection effects by comparing the microbial situation on the surfaces of the site before and after implementing protective measures [41]. Savković Ž. et al. have found that soil particles were often embedded in microbial filaments on the surfaces of ancient Roman stele in Serbia by in situ observation with a portable microscope [43]. A similar situation has been found at the Dadiwan F901 site. Therefore, this indicates that microorganisms have covered the entire environment of the site museum, including the cement building wall (Figure 6B) and the glass curtain wall as well (Figure 6G). The growth of plentiful microorganisms on the surfaces of cement walls was probably caused by the provision of nutrients which originated from oil materials on the surface of molds for pouring concrete. Areas of microbial plaques were mainly distributed on the surface of the glass curtain wall where fingerprints were visible. It was speculated that sweat and lipids were inadvertently left on the glass by field workers as nutrients supplied to microorganisms. Therefore, exogenous materials, especially organic materials, possibly provide nutrients for microbial growth.

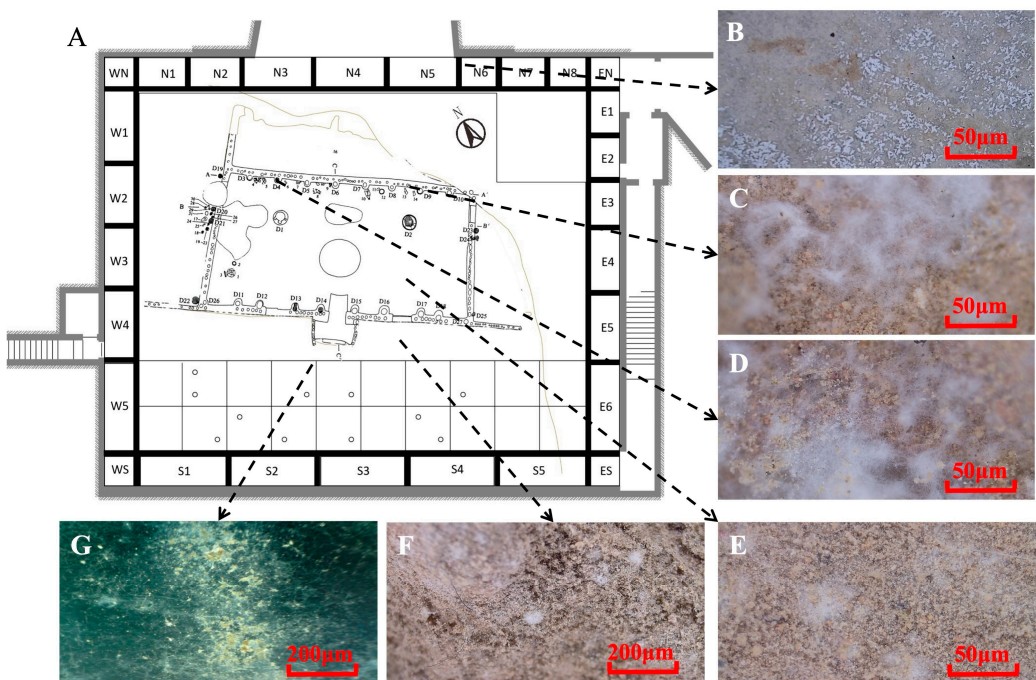

**Figure 6.** Distribution of microorganisms in Dadiwan F901 site. (**A**) Panoramic ichnography of Dadiwan F901 site; (**B–G**) microbial growths on different surfaces of the site observed by portable microscope.

### 3.2.2. Morphologic Characteristics of Residues' Profile Structures

A polarizing microscope was utilized to observe microbial growth and the interaction between microbial filaments and the site soil. Microbial filaments were observed on the surfaces or inside the pores of residues collected from surfaces of the exterior walls of pillar holes (Figure 7A), particles on the ground (Figure 7B), and the cement particles of peripheral buildings around the site (Figure 7C). These filaments displayed a gray color with 50 µm to 400 µm thickness on different surfaces. Additionally, it was also investigated whether microbial filaments stretching into cracks of the samples on the pillar holes could easily lead to the destruction of the site.

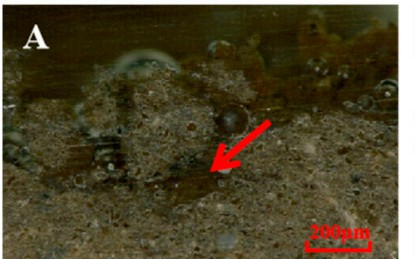
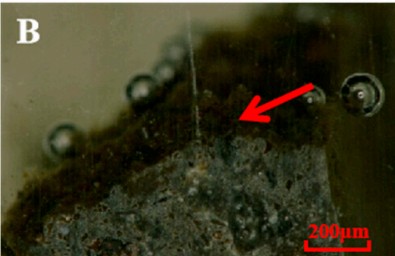
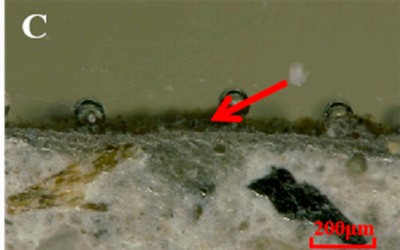

**Figure 7.** The growth situation of harmful microorganisms from different locations at site F901 by polarizing microscope. (**A**) Pillar holes, (**B**) ground, and (**C**) cement residues from peripheral buildings. The red arrows point to the harmful microorganisms.

### 3.3. Evaluation of Prevention Effectiveness against Harmful Microorganisms

#### 3.3.1. Removal Tools and Instructions

After testing in situ, four different lengths of man-made fiber brushes were chosen for clearing away microorganisms and dust particles (Figure 8A). An approximately 4.6 cm contour brush was employed for the first sweep. The brush could effectively remove loose microbial filaments and dust particles on the surface due to its soft and resilient properties. Microbial colonies that were difficult to remove could be eliminated using a cheek brush as

it is as soft as but more resilient than a contour brush. Both the 0.9 cm circular brush and blade brush were effective in removing microorganisms closely attached to the surfaces of the site. Additionally, according to the surface characteristics of the clean area, the circular brush was used on smooth surfaces, while the blade brush was used on coarse surfaces. To prevent the extensive diffusion of microbial filaments and spores to the surrounding environment, a mechanical removal method was designed by combining the brush with a sterilized vacuum device (Figure 8B).

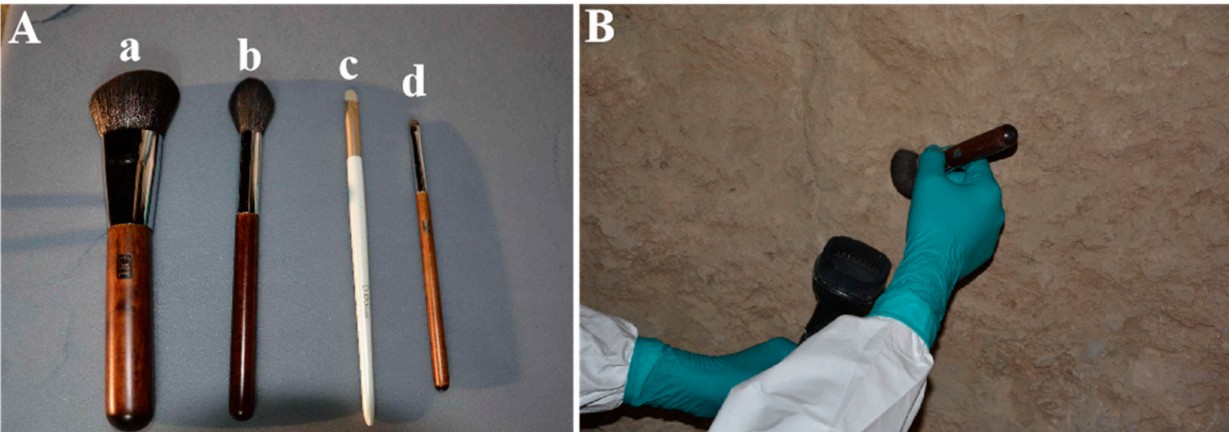

**Figure 8.** Mechanical removal tools (**A**) and usage (**B**). Tools a to d are contour brush, cheek brush, circular brush, and blade brush, respectively.

### 3.3.2. Effectiveness Evaluation of Mechanical Removal and Biocide Inhibition

Figure 9 shows the topographic characteristics of coarse and smooth surfaces of the site wall (A and B), the site ground (C), and the peripheral cement wall (D). Microscopic investigation revealed a significant decrease in microbial filaments after using brushes on the surfaces of these positions, compared to the initial state (see Figure 9A1–D1). However, a small number of residual filaments were observed in cracks and depressions (Figure 9A2–D2). Following two rounds of continuous biocide spray, no obvious microbial filaments were observed on the site surfaces. The microscopic topographies showed no differences when using different biocides or varying concentrations of the same biocide (Figure 9A3–D3).

Microscopical observation showed that mechanical removal had a better effect on smooth and flat surfaces, such as smooth surfaces of the site and cement wall surfaces of peripheral buildings, where no obvious microbial filaments were present due to a stable structure. However, for cracks and depressions on the coarse surfaces and ground of sites with a loose structure, it is necessary to be careful to control the strength used when using a soft brush to remove microbial filaments on the surfaces. The selection of brushes with different shapes was based on the surfaces of the site and proximity to microorganisms. Earthen sites have been subjected to natural and artificial influences for a long time, and some surfaces have different degrees of destruction, cracks, and erosion, which mainly lead to the looseness of earthen site structures. In terms of the situation, any mechanical force has the potential to damage structures, so the selection of tools is an important process for mechanical removal [45–47]. The microscopical results suggested that mechanical removal could hardly effectively remove microorganisms causing deterioration on the site surfaces, and it is necessary to use relevant biocides to compensate for the shortcomings of mechanical removal; biocides can cover the entire surface of the site, including cracks and depressions, and penetrate a certain depth. Presently, the results indicated that the combination of the two methods could more effectively control microbial deterioration on the surfaces at the Dadiwan F901 site.

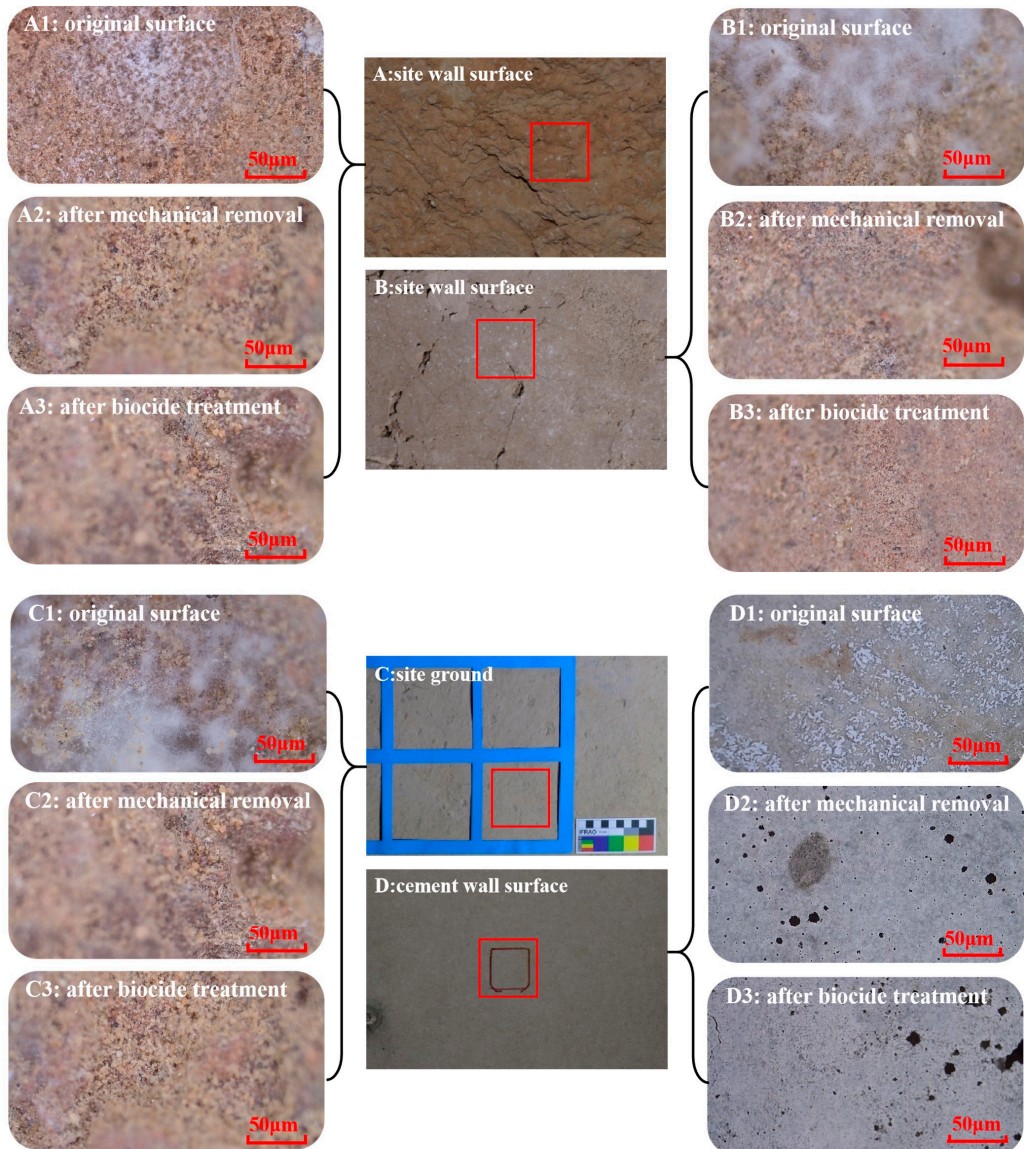

**Figure 9.** Removal effectiveness of harmful microorganisms growing on the surfaces of site walls and peripheral cement walls. (**A–D**) show the surfaces of samples in this study. The red boxes mean the eliminated area; (**A1–D1**) exhibit microcosmic situation of the original surfaces with microorganisms before treatment; (**A2–D2**) and (**A3–D3**) display microcosmic topographies after physical and chemical treatments, respectively.

To examine the elimination effectiveness of microbial deterioration in the site further, both ATP bioluminescence detection and color change difference were utilized for evaluating the efficiency of mechanical removal and biocide application.

ATP bioluminescence assay operates on the principle that ATP in living microorganisms reacts with fluorescein catalyzed by luciferase in a swab tube to release fluorescence, and the number of photons is captured by the instrument to obtain the ATP fluorescence value. A linear relationship exists between the number of living microorganisms allowing an indirect reflection of the microbial count in the sample [48]. Due to its ease of use, non-destructiveness, and fast operation, this assay has been used for microbial investigation on many building surfaces and materials of sites [27,49–53]. In this study, ATP bioluminescence assay was utilized to indirectly detect the number of microorganisms before and after treatments on the site surfaces, respectively. The results showed that the concentration of active microorganisms on the surfaces of the site had been greatly reduced

with mechanical removal and biocides, but removal efficiency was affected by the degree of coarseness, cracks, and other factors on the surfaces of the site.

Similar to microscopic observation, the elimination effectiveness of mechanical removal, as tested by ATP bioluminescence detection, revealed that mechanical removal had a low impact on the elimination of microbial filaments on coarse surfaces with many cracks and depressions, showing an average eliminated ratio of only 24.30% based on Formula (1), followed by the site ground and smooth surface for which the eliminated ratios are 37.70% and 38.00%, respectively. This is in contrast to the effectiveness observed on the surfaces of stone heritage sites by Mascalchi et al., likely due to material differences [41]. The best-eliminated area by mechanical removal is the cement wall surface, with a ratio of 58.86% on average (Table 1).

**Table 1.** The effectiveness evaluation of mechanical removal at different surfaces in site F901 based on ATP bioluminescence assay and color change difference.

| Evaluation Methods | Coarse Surface | Smooth Surface | Site Ground | Cement Wall |
|---|---|---|---|---|
| Eliminated ratio (%) | 24.30 ± 13.14 | 38.00 ± 17.29 | 37.70 ± 15.08 | 58.86 ± 15.19 |
| Color difference value ($\Delta E^*ab$) | 3.64 ± 1.83 | 2.80 ± 1.47 | 2.86 ± 1.55 | 1.93 ± 1.40 |

As is well known, the five biocides are all broad-spectrum. Therein, MD and OIT have higher inhibition abilities against fungi, while DCOIT can effectively suppress bacterial growth [54]. In contrast, BC and K100 are able to inhibit both fungal and bacterial growth. According to a bacteriostatic zone experiment (Figure 10), the results showed that MD and BC had a lower ability to inhibit the growth of culturable microorganisms from the Dadiwan F901 site. Compared to MD and BC, the OIT/DCOIT compound displayed the highest inhibitory effectiveness against microbial growth, with the largest inhibition zones on an R2A solid medium, followed by OIT, DCOIT, and K100, for which there were no significant differences in the strength of inhibition effectiveness, illustrating that OIT, DCOIT, the OIT/DCOIT compound, and K100 could effectively function against the culturable microorganisms of the Dadiwan F901 site.

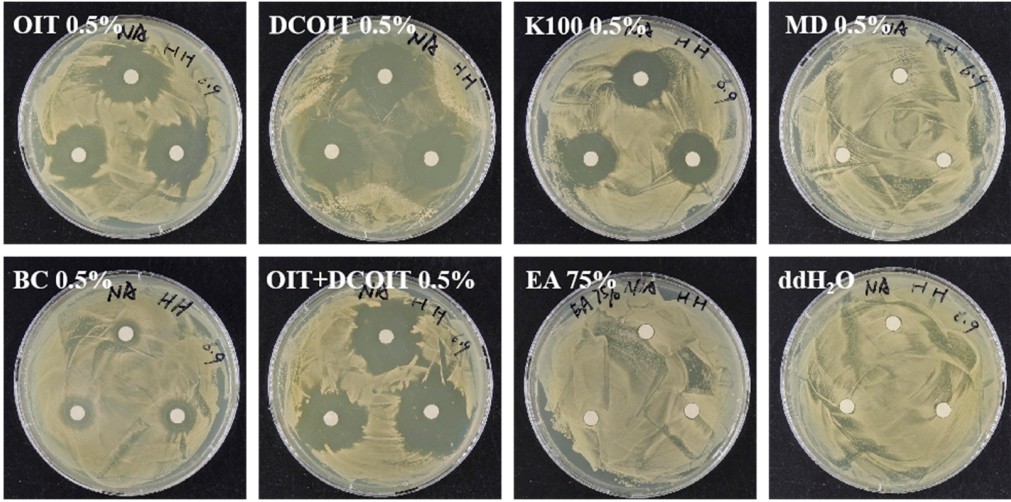

**Figure 10.** Bacteriostatic zone experiment for selection of biocides to eliminate harmful microorganisms at the Dadiwan F901 site.

Regarding harmful microorganisms in the Dadiwan F901 site museum, based on relevant international literature, the standard of low-toxicity, broad-spectrum disinfection, solubility, and identification of the microbial communities of the site samples with

high-throughput sequencing in which genera *Nocardiopsis* and *Saccharopoispora* are dominant bacteria, while the fungal genera of *Penicillium* and *Aspergillus* had higher relative abundance (unpublished data), and the combination of the use of biocides screened by a bacteriostatic zone experiment and the anti-mildew practice of ancient wall paintings [26,27], OIT, DCOIT, K100, and the OIT/DCOIT compound were ultimately confirmed to be used to examine in situ. OIT, DCOIT, and K100, all belonging to isothiazolinone, are efficient and broad-spectrum agents known for inhibiting cell oxygen consumption and enzyme synthesis, oxidizing and destroying cell compounds, etc.; thus, they have been widely applied to prevent microbial deterioration in the preservation of cultural heritage sites. Previous studies have demonstrated the efficacy of OIT in preventing microbial deterioration in the murals of the Majishan Grottoes [26] and the ancient Roman archaeological site of Luni [31]. Additionally, studies of microbial resistance by K100, which was used in murals in the Xu Xianxiu Tomb of Northern Qi Dynasty in Taiyuan [27], a terracotta figurine unearthed in Sun Dayuan Han Tomb in Heze, Shandong [32], the wooden hull of China's "Nanhai No. 1" ancient shipwreck [55], and tomb bricks of the Ding Pottery Han tomb [56], demonstrated that K100 had efficient effect of disinfection of cultural heritage sites.

The elimination effectiveness of OIT, DCOIT, the OIT/DCOIT compound, and K100 was assessed by ATP bioluminescence detection and color change difference. The four types of biocides with four different concentrations could effectively eliminate harmful microorganisms and suppress the microorganisms for one month at least. Figure 8 shows that the OIT/DCOIT compound had highly effective disinfection, and the eliminated ratio could be over 50% compared with the other three biocides after mechanical removal. Because of the different degrees of cracks and depressions on the surfaces or ground of the site over time, the eliminated ratio with various concentrations of the same biocides has significant differences except for the OIT/DCOIT compound, which exhibited almost the highest eliminated ratio compared to the other three biocides in the same concentration. Moreover, the eliminated ratio of the compound could be over 94%, with 97.43% being the highest ratio when a 1% concentration of the compound was used to clear up the microorganisms of the site, making it the most suitable choice for microbial control at the Dadiwan F901 site, followed by OIT and DCOIT, for which the elimination effectiveness of OIT was better on the coarse surface and smooth surface of the site, but lower than the site ground and cement walls of peripheral buildings compared to DCOIT. K100 displayed the lowest elimination effectiveness (Figure 11A–D). This result indicated that K100 was not suitable for microbial control at the Dadiwan F901 site.

The basic criterion for the protection and restoration of cultural relics is to keep their original appearance, which is required to not change from their original state during the protection process. Therein, color change analysis is one of the important criteria to evaluate the reasonability of construction techniques [57] and acts as a portable way to monitor the degree of microbial deterioration on the surfaces of cultural sites [58–62]. Microbial growth could produce a color change in site surfaces, affecting visual appreciation. The carotene produced by microbial colonization appeared rosy on site surfaces in European castles, churches, and underground tombs [63–65]. Moreover, microbial growth could lead to a color change in pigments on site murals [12,66]. Prieto et al. have indicated that color change displayed an increased tendency with microbial coverage and the difference could decrease after dealing with the microorganisms [59]. The current studies reported a criterion for the color difference, which is that when the color difference value ($\Delta E^*_{ab}$) is over 3, this indicates that the color difference is large and the change could be distinguished by the naked eye, while the value is slight and not easy to perceive with values in the range of 1.5 to 3. When the color difference value is lower than 1.5, it is hard to observe a color change by the naked eye [67,68]. However, there are still some controversies about the range of acceptable color difference values from different researchers. Huang and Camerini et al. suggested that the acceptable color difference value ($\Delta E^*ab$) should not be greater than 3 [69,70], but Zhang thought that it was acceptable that the value was lower than 6 [71].

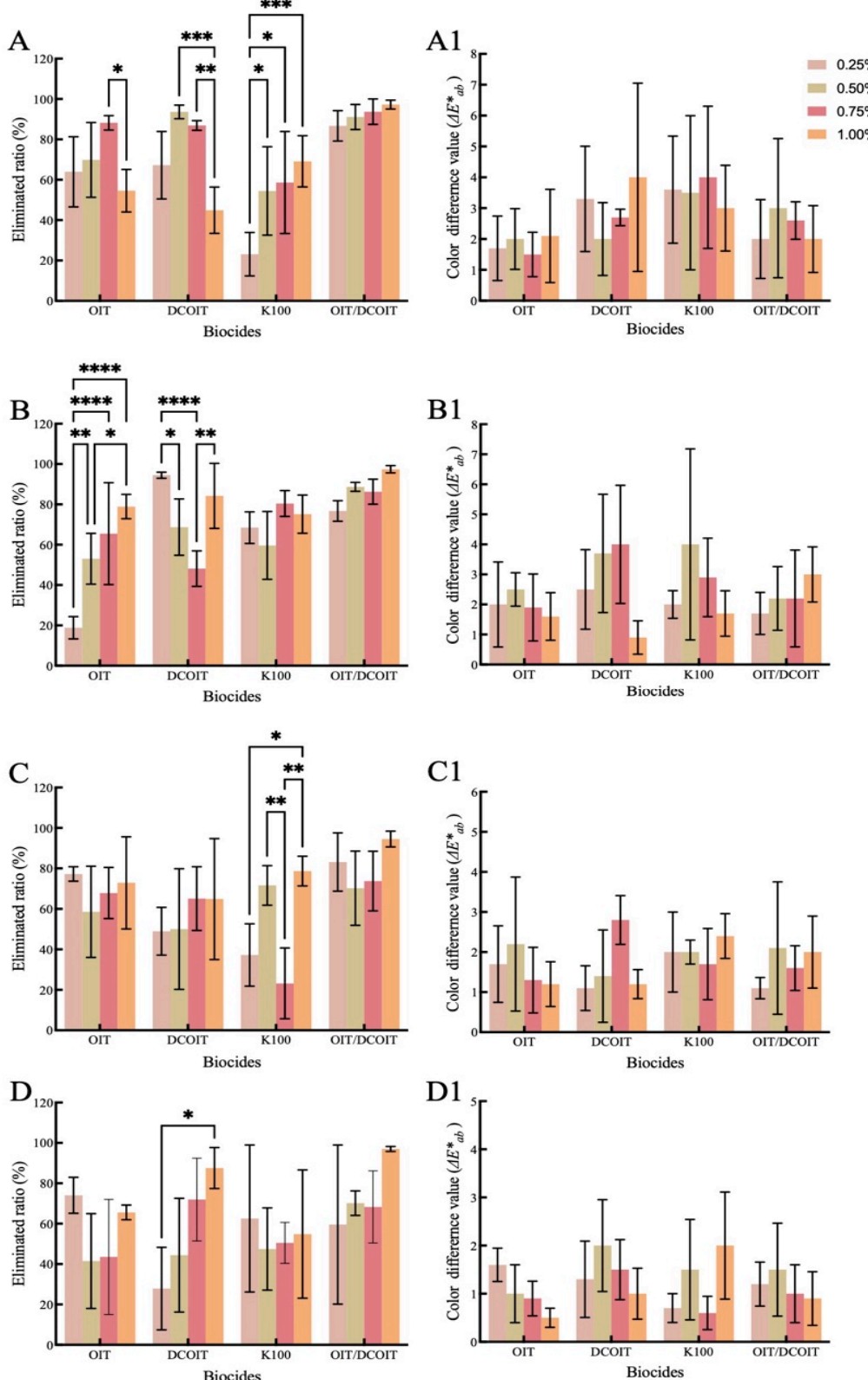

**Figure 11.** Microbial inhibitory effectiveness by ATP bioluminescence detection (**A**–**D**) and color change difference (**A1**–**D1**) treated by different concentrations of four biocides evaluated after mechanical removal on the coarse (**A**,**A1**) and smooth surfaces of site walls (**B**,**B1**), the surfaces of pillar holes (**C**,**C1**), and the cement surfaces of peripheral buildings (**D**,**D1**), respectively. Error bars indicate the standard deviation of three replicates. *p*-value < 0.05. Therein, * means *p*-value < 0.05; ** < 0.01; *** < 0.001; **** < 0.0001.

In the study, it was discovered that the color difference value $\Delta E^*_{ab}$ using Formula (3) at different sites in the test area did not exceed 5; the highest color change was on the coarse wall of the site, with a value of 4.9 after using a 0.75% concentration of K100 (Figure 11A1), and the color on the surface of the cement wall showed the lowest change with 1% OIT, with a value of 0.5 (Figure 11D1). Compared to the spray of biocides, mechanical removal resulted in a more apparent color change, because the colors of microbial plaques on the surfaces of the site are different from the color of the site wall after mechanical removal (Table 1). For the different concentrations of the four biocides, the biocide OIT produced the lowest change in color regardless of concentration, followed by the OIT/DCOIT compound. K100 had the highest effect on the color difference. It was also found that the color difference values on the coarse wall surfaces of the site were larger than the values for the smooth surfaces. Due to the coarse surfaces of the site wall, the colorimeter inevitably produced more systematic errors during the measurement and resulted in large differences in the difference values of chromatism. Therefore, the effects on the color differences of the four biocides are within the acceptable range of less than 6 according to the principles for cultural heritage conservation and without prominent influence on the sense of vision [72], so all four biocides are suitable to eliminate harmful microorganisms in the F901 site.

In conclusion, as to microorganisms causing deterioration at the Dadiwan F901 site, it is suggested that OIT/DCOIT with 1% concentration is the most suitable choice for the prevention of microbial deterioration based on the efficiency of microbial inhibition and color change differences on the surfaces of the site.

### 3.4. Study of Microbial Inhibition by UV Light

### 3.4.1. Eliminated Effectiveness of Culturable Microorganisms under UV Light

Ultraviolet sterilization as one of the physical sterilization methods is characterized by its simplicity of operation, high spectral efficiency, and low secondary pollution and has been commonly used to eliminate microorganisms in the food and medical industry but is rarely used in the conservation of cultural relics [73–75]. With the rolling out of novel ultraviolet light sources, it is gradually gaining popularity as one of the methods for eliminating harmful microorganisms on the surfaces of cultural sites [76,77]. The principle of ultraviolet sterilization involves the destruction and alteration of the molecular structure of DNA in microbial cells by UV irradiation so that the microbial cells die or cannot reproduce to achieve the effect of sterilization [78].

UV light has proven to be effective in suppressing the growth of culturable microorganisms from the Dadiwan F901 site. The effect of inhibition under UV light showed an increased tendency with an extension in irradiation time. There was no microbial colony on the R2A and PDA medium after irradiating for 60 min and 120 min, respectively (Figure 12). Therefore, UV light has significant potential for the long-term prevention of microbial growth in the preservation environment of the Dadiwan F901 site museum.

### 3.4.2. Effect of UV Light on Harmful Microorganisms In Situ

A portable microscope was used to observe the growth situation of microorganisms causing deterioration before and after UV light treatment, as shown in Figure 13. The microbial filaments were fluffy on the surface and extended to the surroundings (Figure 13A). After irradiating for 60 min by UV light, the filaments were wizened (Figure 13A1). A similar phenomenon of microbial filaments treated by UV light appeared on the cement wall surfaces of peripheral buildings outside the site. The raised microbial colonies collapsed and appeared faviform with the inhibition of microbial activities (Figure 13B,B1). This result illustrated that UV light irradiation could affect microbial growth on the surfaces of the Dadiwan F901 site.

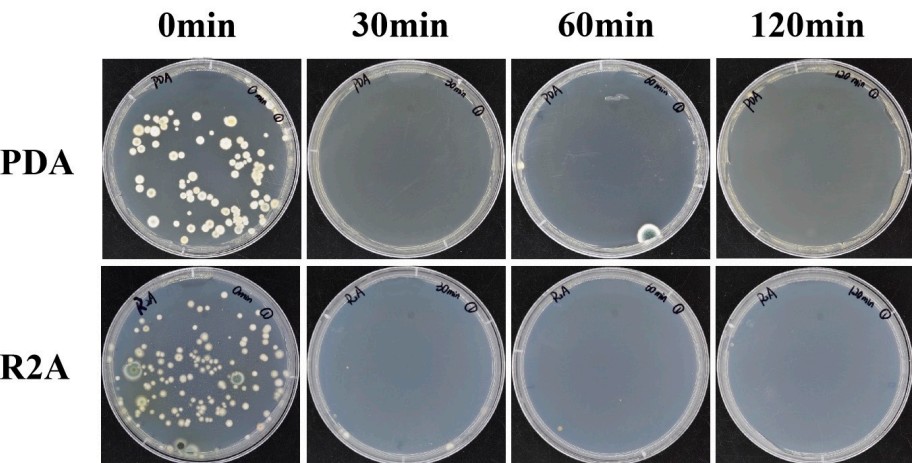

**Figure 12.** The growth situation of culturable microorganisms at different irradiation times in R2A and PDA solid medium, respectively.

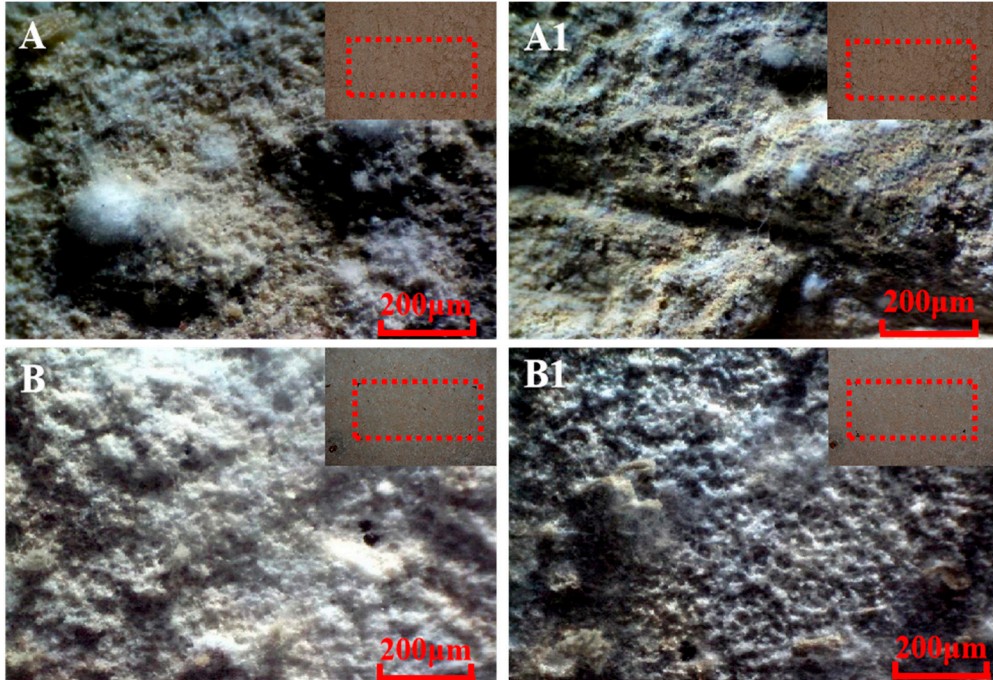

**Figure 13.** Comparison of microscopic morphologies to the harmful microorganisms in site F901 before and after treatment by UV light. (**A**,**A1**) show the microbial morphologies before and after treatment on the wall surface of the site. (**B**,**B1**) display the microbial characteristics before and after treatment on the cement wall surface of peripheral buildings outside the site.

### 3.4.3. Effectiveness Evaluation of UV Light

The ATP fluorescence value on the surfaces of the site gradually reduced with an increase in irradiation time by UV light, as shown in Figure 14. This indicates that UV light caused prominent inhibition of the detrimental microorganisms growing on the site surfaces, since the microbial activity exhibited a tendency to decrease. The value measured on the 10th day after irradiating for 180 min decreased to 1/10 of the original value at 0 min. This result revealed that microbial inhibition by UV light could last for a period of time, while the effect of UV light on the microorganisms is limited because the value could not decrease to a safe value (≤3000 RLU) [79], which might be related to the ultraviolet penetration ability. The disinfection effect on the surface of the cement wall outside the site irradiated by UV light was essentially the same as that on the site. This is because

the biofilm produced by microbial filaments will block the effective penetration of UV to deeper layers, preventing UV light from acting deep in microbial colonies. Previous studies have shown that the disinfectant effects of UV light were affected by installation positions, effective distance, and exposure time [80–85]. It has been reported that viruses show the lowest resistance to UV light, followed by bacteria, and disinfection of fungi exposed to UV light requires much more time compared to bacteria [86–88]. In the Dadiwan F901 site museum, it has been identified that bacteria are dominant microorganisms, indicating that UV light has a significant inhibitory effect in daily life. Additionally, an increase in the layout density of UV light to guarantee the cover of the ultraviolet rays to the site in the Dadiwan F901 site museum could improve disinfective efficiency with an increase in ultraviolet time and indoor air ventilation. Therefore, in the later preventive protection of sites, multiple factors such as irradiation intensity, irradiation time, and possible effects on organic matter cultural relics need to be comprehensively considered to determine the appropriate installation location [89]. Moreover, in the conservation and restoration of cultural heritage sites, it is still necessary to provide mechanical removal and biocide treatment, which can be used together to achieve the ideal effect for microbial control.

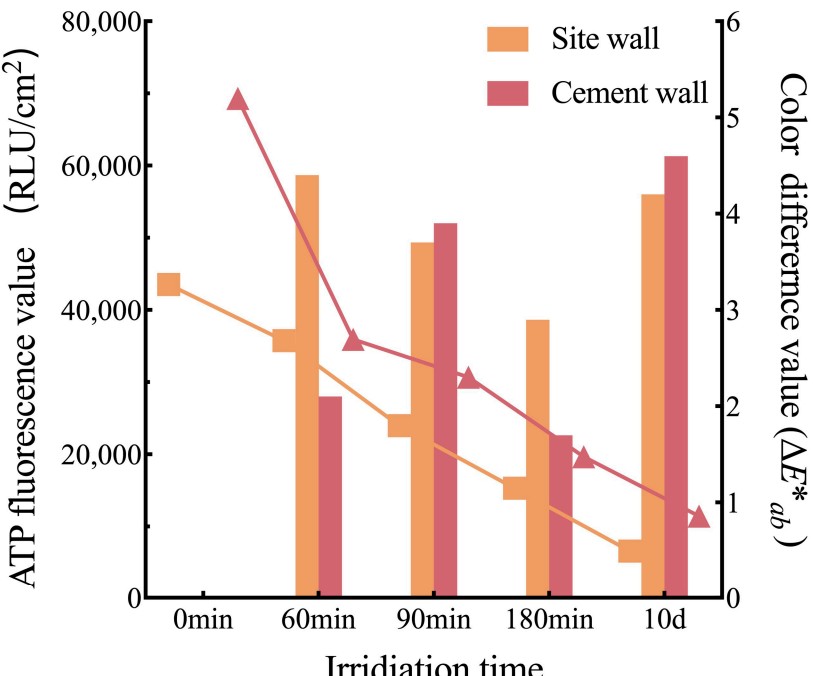

**Figure 14.** Variation in ATP fluorescence value (line chart) and color difference value (column chart) with time in the area irradiated by UV light.

Using UV irradiation, the values of the color difference ($\Delta E^*ab$) on the wall surfaces of the site and peripheral buildings are both lower than 5, without obvious color change (Figure 14). Because the microbial cover produced a significant effect on the color of the cement wall, the change of microbial plaque caused an increased tendency for cement chroma. When the cement wall was irradiated for 180 min, the color difference value was 4.6 after 10 days, but this was still lower than 5 and without obvious change to the naked eye. At the same time, in contrast to the cement wall surface, cracks and depressions on the site surface generally existed. The existence of these could produce larger systemic errors in the measurement of chroma, so the color difference had a greater change in its acceptable range.

The data suggested that UV light could be regarded as a preventive measure for the protection of the Dadiwan F901 site. It is the first time that the effect of ultraviolet sterilization on an earthen site and the killing of microorganisms for a long time has been explored as a routine sterilization measure.

## 4. Conclusions

Due to changes in the preservation environment of the Dadiwan F901 site and the influence of high temperatures and humidity after the closure after the installation of glass and the construction of a new building, extensive growth of microbial filaments appeared on the site surfaces. Therefore, it is urgent to find some scientific protection methods to control microbial activities, which may support the sustainable preservation and ability of the public to visit this site. This study employed physical and chemical methods to eliminate the microorganisms that caused deterioration in a large-scale project and systematically evaluated the effectiveness of treatments using mechanical tools, types and concentrations of biocides, and irradiation time of ultraviolet sterilization by non-destructive methods such as a portable microscope, ATP bioluminescence assay, and color difference analysis. The results provide scientific data to support conservation techniques and the optimization of process screening for microbial control in earthen sites. The results indicate that the main factor affecting the mechanical removal effect was the smoothness of the wall surface. The clearance rate of the cement wall was the highest (73.62%), but the clearance rate of the thick wall of the site was the lowest at 8.44%. Among the four types of biocide, the best elimination effectiveness was by using the OIT/DCOIT compound biocide, with an elimination rate as high as 97.43%, followed by DCOIT and OIT, while K100 had a poor effect, with a minimum value of only 23.13%. In addition, the biocide K100 had a greater impact on the color difference of the site, so K100 was not used at this earthen site. Although the sterilization of UV light has a limited ability for the site due to the coarse surfaces of the site wall and spatial range, it has long-term effectiveness in microbial growth inhibition. Consequently, comprehensive cover and regular irradiation of ultraviolet rays in the site museum could efficiently prevent microbial growth while maintaining indoor air ventilation. The results of the examination provide scientific data to support conservation techniques and optimize process screening for microbial control in large-scale projects of earthen sites. The development of related protection techniques will be beneficial for the long-term preservation and sustainable utilization of similar archeological site museums.

**Author Contributions:** Conceptualization, F.W.; methodology, R.X. and D.H.; investigation, R.X. and H.Z.; resources, F.W. and G.Z.; writing—original draft, R.X. and Y.C.; writing—review and editing, Y.C. and F.W.; supervision, F.W. and G.Z.; project administration, F.W., Q.L. and G.Z.; and funding acquisition, F.W. and G.Z. All authors have read and agreed to the published version of the manuscript.

**Funding:** This research was funded by the National Natural Science Foundation of China, grant no. 32060258, the Open Project of The Museum of the Terracotta and Horses of Qin Shihuang, grant no. Qkfkt202106, the Project of Gansu Provincial Bureau of Cultural Heritage (nos. GWJ202011; GSWW202227).

**Institutional Review Board Statement:** Not applicable.

**Informed Consent Statement:** Not applicable.

**Data Availability Statement:** Data are contained within the article.

**Acknowledgments:** The authors would like to give thanks for the support and help of the Dadiwan Site Museum's workers.

**Conflicts of Interest:** The authors declare no conflicts of interest.

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
