# Peer review of "Preliminary Study on Microbial Deterioration Control and Effectiveness Evaluation in the Neolithic Prehistoric Archaeological Site of Dadiwan, Northwest China"

_coatings, doi:10.3390/coatings14010100_

Round 1

Reviewer 1 Report

Comments and Suggestions for Authors

A very interesting manuscript. It seems to me that the results obtained by the authors constitute another step in research on preventing deterioration and destruction of important monuments and securing cultural heritage. Therefore, I think it is worth improving some elements of the text.

In accordance with the "Instructions for Authors", the Coatings journal suggests separate sections: "Results" and "Discussion", but in many volumes I found a combination of these sections, as in the reviewed manuscript. However, in my opinion it would be much clearer to separate these sections. In its current form, the Discussion seems a bit too limited. However, I leave this suggestion to the Editor's discretion.

In my opinion, there is a bit too much information about other archaeological sites in the Introduction. I think it is worth shortening the Introduction a bit and moving these elements to the Discussion. However, the Discussion is a bit too poor for me. There are too few references to applied/experimental deterioration prevention techniques. I propose to expand a bit on how the destruction of ancient buildings/tombs is currently being carried out.

I regret that the authors did not perform a microbiological analysis to identify the microorganisms responsible for deterioration. This would be a very interesting element of this work. Therefore, I am very interested in why/on what basis the media: PDA and R2A were used to assess UV activity? Please explain this in the Methodology.

The authors use the term "pathogenic microorganisms" several times. However, since the microorganisms have not been identified, it cannot be assessed whether they are pathogenic. By definition, pathogenic means causing disease. In this case, the research concerned microorganisms causing deterioration/destruction of abiotic structures, so we cannot talk about pathogenicity here. I kindly ask the authors to replace this term with another one, e.g. microorganisms causing deterioration.

In the Conclusions, the authors wrote: "It also plays a vital role for the preventive control of environmental airborne microorganisms around earthen sites." However, in their research they dealt with microorganisms found on abiotic surfaces and in the soil. They did not examine the air around the sites. Therefore, they cannot conclude about airborne microorganisms. In my opinion, this conclusion is too broad, considering the research done.

One final technical note: References does not provide complete data. In the case of multiple references, not all authors are provided (according to the Instructions for Authors, a complete list should be provided), instead after 3 authors there is "etc". Additionally, some references are not formatted according to the recommendations, e.g. reference 18 or 70. Please edit all references according to the recommendations.

Reviewer 2 Report

Comments and Suggestions for Authors

Some corrections in text

Recheck references list 

Reviewer 3 Report

Comments and Suggestions for Authors

Report

Manuscript title:

Microbial Deterioration Control and Effectiveness Evaluation in the Neolithic Prehistoric Archaeological Site of Dadiwan, Northwest China

I thank the authors for their great efforts in this manuscript. Some comments are as follows:

Title

Introduction

In the introduction section, the authors explain the research problem of microbial deterioration of some famous archaeological sites and mention some previous studies related to their research problem. However, they did not talk about previous work on controlling microbial deterioration in similar studies to show the novelty of their research and the difference from others. Accordingly, it is necessary to refer to some previous studies on materials and methods of controlling microbial deterioration in some archaeological sites.

Materials and methods

· There is a general note in the Materials and Methods section, as there are no references in all sections except Section 2.3. Removal methods of pathogenic microorganisms.

In section 2.1. (study site), the authors referred to the annual average temperatures and relative humidity in the site studied, but they did not write the reference that mentioned this or the method of measurement that was made.

In section 2.2.2. (Profile characteristics of samples), Does the method of polishing Small particle residues attached to the obvious microbial colonies for examination with a polarizing microscope not affect microbial growth? I kindly ask the authors to write more details about the polishing process of the samples taken.

In section 2.3. (Removal methods of pathogenic microorganisms), On page 3, lines 119-120, the word restoration should be changed to preservation.

How did the authors know that the organisms were pathogenic? Did they run the test for that? If the answer is yes, this test should be written in the materials and methods section.

In section 2.3.2. (In-situ screen of chemical biocides), the authors should mention the criteria based on which the OIT/DCOIT compound was selected from a conservation point of view. They should write in brief the chemical and physical properties of this compound. They also should write the previous studies done on this compound in the conservation field. International charters and conventions prohibit conducting experimental studies on buildings and archaeological artifacts. It is necessary to create simulation models of archaeological objects and experiment on them. Is it correct that the authors conducted their experimental studies on some parts (even if they were very small) of the original site studied? They should provide reasons why the experimental study was carried out on some parts of the original site. Authors should write their reasons for choosing the concentrations of 0.5%, 0.50%, 0.75%, and 1.0%.

In section 2.4.1. (Inhibition of UV light to culturable microorganisms), this point is not clear for readers. I kindly ask the authors some questions concerning this point:

-   Why did the authors not identify the isolated fungi and bacteria?

-   Why did they not study its enzymatic activity? By studying enzymatic activity, they can detect the most dangerous fungi and bacteria, which can be used in their experimental studies using OIT/DCOIT compound or ultraviolet radiation.

-   Why the duration of exposure to ultraviolet radiation did on cultures of isolated fungal and bacterial (0min, 30min, 60min, 120min) differ from exposure to parts of the studied site (0min, 60min, 90min, and 180min)?

-   Why did the authors choose sterilization with ultraviolet radiation, especially since sterilization takes place in an open environment, and sterilization with ultraviolet radiation does not provide long-term sterilization and protection, especially in the conditions of the studied site, whose average annual relative humidity was 73.6%?

-   Why did the authors use two methods for inhibition of microorganisms: chemical disinfection using OIT/DCOIT compound, and physical sterilization using ultraviolet radiation?

·      In section 2.5.2. (ATP bioluminescence assay): the equation mentioned in this section Eliminated ratio (%) = [(C0-C)/C0] × 100%, and the author identified that the value C0 referred to the treatment by mechanical removal, and at the same time they identified that the value C also referred to the treatment by mechanical removal. Is it true? 

Results and discussion

· In section 3.1.1. (In-situ morphology and distribution of pathogenic microorganisms), the authors mentioned in the first paragraph some relevant studies. It is better to transfer this paragraph to the introduction section.

· In the same section (3.1.1.) the authors said that rarely studies have been done concerning the control and preventive measurements. Many studies have been conducted on controlling and preventing microbial damage in archaeological and heritage sites, including but not limited to the following:

-   Cappitelli, F., Cattò, C., Villa, F., The Control of Cultural Heritage Microbial Deterioration, Microorganisms, Vol. 8, 1542, 2020, pp. 1-20.

-   Sun, M.,  Zhang, F., Huang, X., Han, Y., Jiang, N., Cui, B., Guo, Q., Kong, M., Song, L., Pan, J., Analysis of Microbial Community in the Archaeological Ruins of Liangzhu City and Study on Protective Materials, Frontiers in microbiology, Vol. 11, article 684, 2020, pp. 1-14.

-   Wang, Y., Huang, W., Han, Y., Huang, X., Wang, C., Ma, K., Kong, M., Jiang, N., Pan, J., Microbial diversity of archaeological ruins of Liangzhu City and its correlation with environmental factors, International Biodeterioration & Biodegradation, Vol. 175, 2022, 105501.

-   Khalil, M. M. E., Mekawey, A. A. I., Alatawi, F. A., Microbial Deterioration of the Archaeological Nujoumi Dome (Egypt-Aswan): Identification and Suggested Control Treatments by Natural Products, Journal of Pure and Applied Microbiology, Vol. 16 (2), article No. 7377, 2022, pp. 990-1003.

-   Joseph, E., Microorganisms in the Deterioration and Preservation of Cultural Heritage, Springer, e-book, 2021.

· Page 5, line 228, the word restoration should be changed to preservation.

· Page 8, line 301, Microscopical observation suggested… should be changed to Microscopical observation showed ……

· Page 8, line 312, The microscopical results suggested … should be changed to The microscopical results stated that ……

· Page 10, line 359, the word conservation should change to Preservation.

· Section 3.3.3. (Effectiveness evaluation of UV light): I have a question for the authors: Does ultraviolet sterilization provide high efficiency in the inhibition of microorganisms in open areas such as the studied site?

Conclusion

· The conclusions were very poor and were more like the abstract. It did not talk about what was concluded from the evaluated materials and methods, especially biocide and ultraviolet sterilization, the differences between them, and what the authors could recommend in this regard.

· Authors should reconsider writing conclusions so that they are convincing to the reader, especially the specialized reader.

Recommendation

I kindly recommend a major revision of this manuscript if the authors make all the comments mentioned.

Comments on the Quality of English Language

Reviewer 4 Report

Comments and Suggestions for Authors

Dear Authors

I have carefully reviewed the article entitled "Microbial Deterioration Control and Effectiveness Evaluation in the Neolithic Prehistoric Archaeological Site of Dadiwan, Northwest China." While the paper is well-written, I would like to express some concerns regarding its scientific contributions.

The overall structure of the paper is commendable; however, it lacks novelty in terms of scientific findings and knowledge addition. The techniques employed are rather basic and widely recognized, with no evident incorporation of innovative approaches. The paper predominantly relies on standard operating procedures (SOPs) that are already established in the field, without introducing any new or advanced scientific methodologies.

Moreover, the microbiological aspect of the paper is deemed basic, lacking a critical analysis of the subject matter. The observations presented appear to be evident and predictable, as the authors have predominantly utilized well-established methods with proven outcomes. This approach, unfortunately, diminishes the perceived depth and scientific rigor of the research.

I appreciate the effort put into this work and encourage the authors to consider revisiting the study with a focus on incorporating innovative approaches and providing a more critical analysis of the microbiological aspects.

In light of these concerns, the results come across as routine observations rather than the outcome of a comprehensive scientific investigation. The absence of novel insights and advanced methodologies raises questions about the overall scientific merit of the paper. Consequently, I am inclined to recommend rejection based on these grounds.

Round 2

Reviewer 3 Report

Comments and Suggestions for Authors

I thank the authors for their efforts in the modification of their manuscript. I have two comments as follows:

- the author must identify the microbes (fungi and bacteria) at least by morphological identification method.

- The title of the manuscript must change to:

"Preliminary Study on Microbial Deterioration Control and Effectiveness Evaluation in the Neolithic Prehistoric Archaeological Site of Dadiwan, Northwest China"

Comments on the Quality of English Language

For editor

Dear Editor,

I kindly inform you that the two comments that were written to authors must be done before your acceptance.

Best regards 

Reviewer 4 Report

Comments and Suggestions for Authors

Authors have made few updates in the manuscript therefore i would like to accept the manuscript in the current form

Round 3

Reviewer 3 Report

Comments and Suggestions for Authors

The English language can be improved.

Comments on the Quality of English Language

The English language can be improved.